# Severe Asthma, Telemedicine, and Self-Administered Therapy: Listening First to the Patient

**DOI:** 10.3390/jcm11040960

**Published:** 2022-02-12

**Authors:** Gabriella Guarnieri, Marco Caminati, Alessia Achille, Rachele Vaia, Fulvia Chieco Bianchi, Gianenrico Senna, Andrea Vianello

**Affiliations:** 1Respiratory Pathophysiology Unit, Department of Cardiac, Thoracic, Vascular Sciences and Public Health, University of Padova, Via Giustiniani 2, 35126 Padova, Italy; alessia.achille@aopd.veneto.it (A.A.); fulvia.chiecobianchi@aopd.veneto.it (F.C.B.); andrea.vianello.1@unipd.it (A.V.); 2Department of Medicine, University of Verona, Verona University Hospital, P.le L. Scuro 10, 37134 Verona, Italy; marco.caminati@univr.it (M.C.); gianenrico.senna@univr.it (G.S.); 3Allergy Unit, Asthma Center, Verona University Hospital, P.le L. Scuro 10, 37134 Verona, Italy; rachele.vaia@univr.it

**Keywords:** patient empowerment, home-administration, patient preferences, adherence, biologics

## Abstract

Severe asthma patients are at an increased risk of major complications and they need to be monitored regularly. The COVID-19 pandemic has notably impacted on the health care resources. The telemedicine approach applied to the follow-up of asthmatic patients has been proven to be effective in monitoring their disease and their adherence to the therapy. The aim of our study was to investigate the satisfaction of severe asthma patients before the activation of a telemedicine management, as well as their current experience with self-administration of injection therapy. An *ad hoc* questionnaire was developed and sent by e-mail to 180 severe asthma patients. Most of subjects, 82%, were confident with the idea of doing self-measurements and self-managing their disease. Further, 77% of subjects favoured to carry out virtual visits and telemedicine. Regarding the home treatment, 93% of patients considered the self-injection therapy easy, 94% of subjects felt safe, and 93% were not worried while self-administering. Only mild adverse events were reported in 22% of patients after self-administration. Our results showed an agreement between what is considered necessary and practicable by healthcare personnel and what is perceived by the severe asthma patients in terms of treatment and monitoring of the disease with Telehealth. Biologics have a safety profile and can be easily self-administred at home.

## 1. Introduction

Severe asthma patients are at an increased risk of major and frequent complications, including emergency room and hospital admissions; therefore, they need appropriate pharmacological treatment to be tapered according to disease control [1,2]. The COVID-19 pandemic has notably impacted health care resources, and the restrictions on in person-visits have in turn widely promoted the use of telemedicine for several chronic conditions, including bronchial asthma [3]. According to the results of previous studies, the telemedicine approach applied to the follow-up of asthmatic patients has been proven to be effective in reducing respiratory symptoms and improving quality of life [4,5]. Another study described a similar disease control in patients managed using telemedicine or by the conventional face-to-face visits, suggesting telemedicine is an effective tool for home-monitoring [6]. Telemedicine may be reliable for the management of asthma at any level of severity. In fact, besides the home spirometry and digital visits, the adherence to therapy can also be easily monitored [7]. Severe asthma patients were regularly telemonitored for their home self-administered biologic therapy [8]. However, the recent development of inhaler trackers assessing the real-time usage of inhaled drugs may represent a step forward in achieving a better adherence to asthma treatment at any level of severity. The empowering of patients’, which represents a crucial step for achieving an optimal asthma control, can be implemented by e-health technologies and telemedicine, including the availability of digital apps providing warning notifications [9,10]. The patient’s perception and judgement about the management of their disease through telemedicine tools represents an underestimated issue. In fact, telehealth (i.e., virtual consultation) does exclude the physical interaction between patients and health care professionals, thus reducing the usual emotional contact. However, patient satisfaction with self-administration of biologics in severe asthma has been reported [11,12].

The aim of our study was to preliminarily explore, in a group of severe asthma patients, their aptitudes and satisfaction towards telemedicine tools before the activation of a telemedicine management complemented by a standard of care, and to investigate their experience on the self-administration of injection therapy.

## 2. Methods

An *ad hoc* questionnaire was developed by the physicians operating in the severe asthma referral centres participating in the study (Padua and Verona, northeast of Italy) and sent by e-mail to the patients in order to investigate the aptitudes and satisfaction of telemedicine before its activation as a complementary standard patient management approach. Our telemedicine program includes the execution of quarterly virtual visits, equipping patients with spirometer, oxyhemoglobin saturation meter, and standardized questionnaires such as Asthma Control Test (ACT) [13] and Asthma Control Questionnaire (ACQ-6) [14]. Patients would be trained in-person at the clinic in advance in order to independently perform the measurement of respiratory volumes, flows, and saturation at home. Alternatively, in regards to the self-administration of injection therapy, patients had already been trained by the physician in the clinic on the procedures to be performed. Therefore, in this study some items were included in the questionnaire in order to investigate the patients’ experience of self-administration therapy at home. The questionnaire consisted of 13 items, with each question rated from 0 to 4, according to grade of satisfaction, where 0 is “extremely”, 1 “very much”, 2 “moderately”, 3 “a little”, and 4 “not at all” satisfied. The first item was regarding the self-assessment of breathing, oxygen saturation, state of health at home, and patient empowerment; the second question was about the patient’s ability with the technology, while the third regarded the proposal to make a virtual visit. The fourth question investigated whether the patient perceived that they were properly followed-up by the doctor with telemedicine management. The fifth explored whether the patient preferred the standard in-person visit instead of a virtual visit. The other eight questions were related to self-injection therapy at home to the evaluation of how easy the injection procedure was (sixth), as well as the syringe or injector (tenth), safety and skill (seventh), feeling unanxious (eighth), and any adverse events after injection (ninth). The eleventh question was whether the patient would recommend the procedure to another severe asthma patient, if training was clear (twelfth), and the lastly it was investigated if the patient’s choice was influenced by the COVID-19 pandemic.

## 3. Results

The questionnaire was sent to 180 severe asthma patients regularly treated by two allergy and respiratory referral centres for severe asthma located in the northeast of Italy (Verona and Padua), on biologic treatment. Overall, 167 subjects (93%) completed the questionnaires. The study population consisted of 54% females, with an average age of 55 ± 13 (mean ± SD) years, under the following biologic treatment: 37% Mepolizumab, 28% Omalizumab, 31% Benralizumab, and 4% Dupilumab. All answers to the questionnaire are summarized in Table 1. Most subjects (82%) declared to be confident with the idea of doing the self-measurements and self-managing their disease (31% extremely), whereas 7% disliked it (Figure 1A).

When asked about their preliminary opinions on virtual and telemedicine visits, 77% of subjects reported that they would be satisfied with them, and that they would feel adequately followed up by doctors; however, 72% expressed a preference for the conventional standard visits in the clinic. In our study most of the patients were average skilled with technology, being only 5% completely unable. Almost all patients (93%) considered it easy enough to complete the self-injection therapy (52% extremely), as well as easy to use the auto-injector device (94%). We did not observe differences in the positive judgment among patients between the syringe and auto-injector therapy, when clustering the patients according to the type of biologic. Most of patients (94%) felt safe (50% extremely, 31% very much, and 13% moderately) and 93% were not worried while self-administering, with only 1% of subjects feeling insecure and anxious. Some symptoms were reported by 22% of patients within the 2–3 days following self-administration (Figure 1B). When requested to detail those symptoms, they included the following: small bruises at the injection site (34%), mild headache in the evening of administration (45%), and in the remaining cases fatigue was reported. Therefore, only mild adverse events were reported, which subsided without therapy. Except for 3% of patients, all would recommend self-administration (54% extremely). The training performed for the self-administration at the clinic was rated as satisfactory by almost all patients (66% extremely, 31 very much, 2% moderately, 1% not at all). The patient’s choice was not influenced by the COVID-19 pandemic in 51% of cases, while in 19% and 13% extremely and very much, respectively.

## 4. Discussion

It is frequently seen that what is considered by physicians to be useful and clinically significant for patients do not always correspond to the actual or perceived need of the patients themselves. This can result in a low therapeutic adherence, or a poor professional-patient relationship [15]. During the COVID-19 pandemic, every effort was carried out to prevent a possible infection in patients affected by severe chronic respiratory diseases, such as severe asthma. Specific pathways have been identified to allow patients to receive appropriate medical care and to carry on the biological therapies, thus preventing the lack of disease control, as well as acute severe complications [3,4,5,6,7,8,9,10]. Thus far, to our current knowledge, no investigations have been carried out among the severe asthma patients to evaluate their perspectives about the telemedicine approach. Our results showed an agreement between what is considered necessary and practicable by healthcare personnel and what is appreciated and perceived by the patients affected by severe asthma, in terms of treatment and monitoring of the disease with Telehealth. The severe asthma patient is accustomed to frequent medical visits, a great number of drugs (needing to be often modulated in terms of dose and frequency), and experiencing complications. The healthcare personnel-patient relationship is, therefore, very close. However, we found that training for self-administration therapy was effective, and the idea of self-measurements for breathing and conducting virtual visits were highly appreciated by patients and strengthened their personal empowerment. Overall, our patients were in favour of a telemedicine approach without the fear of feeling abandoned. The biologics used for severe asthma were equipped with handy injectors, which were considered by the patients to be “very easy”. Though biological treatments could be considered more “invasive” than tablets, they were self-administered without anxiety by most of our patients. Biologics in severe asthma, as known from the literature, are safe overall [16]. Our data also confirm their safety profile. In fact more than two out of three among our patients did not experience any adverse events after self-administration, and in the other cases they experienced common and mild symptoms. The COVID-19 pandemic had affected almost half of our patient’s choices to join telemedicine and the biologics self-administration. Further, an agreement emerged between healthcare personnel and patients for safety issues, as well as to keep the severe asthmatic patients monitored. Following the overall technological evolution in the field of medicine, it was very likely that a telemedicine revolution would have characterized our way of practicing in a few short years, but it is undeniable that the COVID-19 pandemic significantly quickened this process. The small sample size represents a major limitation of this study. However, patients were representative of the north-east Italian population, considering the low prevalence of severe asthma. Furthermore, a high response rate was observed, which may be explained by the peculiar patient-doctor relationship, much closer than in other contexts.

Another limitation of the study was determined by timing issues, given the ongoing COVID-19 pandemic; in fact, the usual standardization procedures were not followed, therefore, the individual questions were not pre-tested. However, the elective objective of the study was to conduct a survey, and, therefore, based on clinical experience and literature data, the questions were generated, reduced, and formatted. Finally, the questionnaires were completed in the absence of the investigator. This eliminated the possibility of bias from the investigator; however, this could have led to the misunderstanding of certain, not so easily comprehensible, questions.

## 5. Conclusions

Our data, according to the severe asthmatics reports, suggests extending telemedicine in routine clinical practice to other asthmatic patients, as well as to general patients with chronic respiratory diseases. Finally, high treatment satisfaction with one’s medication would encourage adherence in clinical practice. For the patients not satisfied with the telemedicine approach, the benefit of biologics needs to be re-assessed. Further research in the field may include the re-administration of the same questionnaire, to be compared with the previous results, in order to include the telemedicine management as a precision medicine tool.

## Figures and Tables

**Figure 1 jcm-11-00960-f001:**
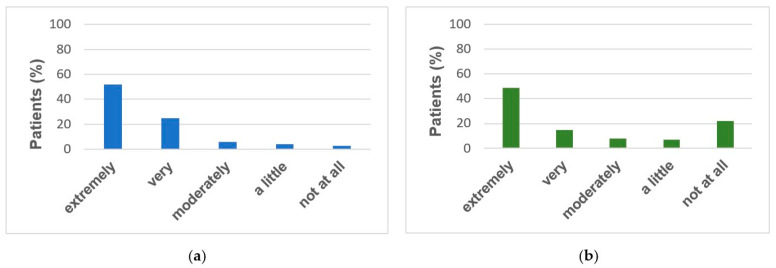
Perspective, Agreement, and Safety of Self-administred Biologics: Answers of Severe Asthma Patients. (**a**) Easy self-injection therapy at home. (**b**) Adverse events after self-injection.

**Table 1 jcm-11-00960-t001:** Patient perception and satisfaction of Telemedicine tools and self-administration of injection therapy.

Items		Extremely (% Patients)	Very Much (% Patients)	Moderately (% Patients)	A Little (% Patients)	Not at All (% Patients)
1	Self-assessment of breathing and oxygen saturation, patient empowerment	31	25	26	11	7
2	Patient’s ability with technology	27	28	28	12	5
3	Satisfation to make a virtual visit	30	27	20	16	7
4	Perception to be properly followed-up by the doctor with telemedicine management	21	24	31	15	9
5	Preference of in-person versus virtual visit	21	28	23	19	9
6	Easy self-injection therapy at home	52	25	16	4	3
7	Safety and skill about the injection procedure	50	31	13	5	1
8	Feeling unanxious	50	25	18	6	1
9	Adverse events after self-injection	49	15	8	7	22
10	Easy syringe or injector	60	27	7	0	6
11	Recommendation to another severe asthma patient about self-injection	54	32	10	1	3
12	Clear training at clinic	66	31	2	0	1
13	Patient’s choice influenced by the COVID-19 pandemic	19	13	17	17	34

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
