# Peer review of "Severe Asthma, Telemedicine, and Self-Administered Therapy: Listening First to the Patient"

_jcm, 2022, doi:10.3390/jcm11040960_

Round 1

Reviewer 1 Report

I think its overall a great concept and study with some issues with methodology. The process of having patients manage their disease process on their own as much as able specially during the time of the pandemic is a great and much required thing, as it has become increasingly difficult to have a fully functional office/ clinic setting during the peaks of a pandemic. This being a survey-based study so is qualitative with not a lot of patients.

The big concern is generalizability- in patient population with limited access of devices, poor education and financial status. I would ask the authors to explain how the questions were generated, reduced and formatted. Were individual questions pre-tested. Also was the entire questionnaire pilot tested. Could the authors comment on how they sought to limit response and non-response bias.

Author Response

Dear Reviewer,

we read your comments carefully. We thank you for the considerations given and methodological suggestions. Please read the clarifications and changes requested directly in the text

Reviewer 2 Report

Thank you for the opportunity to review following Brief Communication. Authors of the manuscript present the results of a survey assessing satisfation of severe asthmatic patients under self-administered biologic treatment. 

Instead of small sample size (according to the authors' declaration the sample is representative for the region) high response rate (93%), what is rather rare for surveys conducted indirectly. 

In my opinion following study presents valuable informations for clinitians taking care of this group of patients however I have a few comments:

1. From my perspective the aim of the study is not clear. As far as I understood the authors' intention the purpose of the study was to evaluate the satisfaction of patients being under biological treatment administered by themselves at home. However there is a lack of information what kind of telemedicine management was carried out.  The statement:  ,,before the activation of a telemedicine management" is not clear.

2. I am not sure whether it can be concluded that: ,,our patients enjoyed telemedicine very much and did not feel abandoned in the context of the telemedicine approach." on the base of received informations from the questionnaire. 

Author Response

Dear Reviewer

we thank you for your very accurate and useful comments to improve our manuscript. Please read the changes requested directly in the text 

Round 2

Reviewer 2 Report

The manusctipt imroved, however it needs minor revision.

In the line 77 please provide full names of questionnaires ACT and ACQ-6, as well as references.

In Methods authors inform that patients were instructed to assess their saturation adn spirometry. Was the training in-person or on-line. There is no information about self-injections: were patients trained in the clinic, or on-line, who conducted the training?

Author Response

Dear Reviewer,

we thank you for your comments and suggestions. Please see the text, we have in fact entered what you requested.